# Continuous Intraoperative Nerve Monitoring of a Non-Recurrent Laryngeal Nerve: Real-Life Data of a High-Volume Thyroid Surgery Center

**DOI:** 10.3390/cancers16051007

**Published:** 2024-02-29

**Authors:** Ann-Kathrin Lederer, Julia I. Staubitz-Vernazza, Rabea Margies, Florian Wild, Thomas J. Musholt

**Affiliations:** 1Section of Endocrine Surgery, Department of General, Visceral and Transplantation Surgery, University Medical Center Mainz, Johannes Gutenberg-University Mainz, 55131 Mainz, Germanyrabea.margies@unimedizin-mainz.de (R.M.);; 2Center for Complementary Medicine, Department of Medicine II, Medical Center-University of Freiburg, Faculty of Medicine, University of Freiburg, 79106 Freiburg, Germany

**Keywords:** continuous intraoperative nerve monitoring, neuromonitoring, IONM, thyroidectomy, neck surgery, vocal cord palsy, non-recurrent laryngeal nerve, arteria lusoria

## Abstract

**Simple Summary:**

This retrospective case series provides real-life data of patients with a non-recurrent laryngeal nerve who underwent surgery at a high-volume thyroid surgery center using continuous intraoperative nerve monitoring to prevent vocal cord palsy. The results emphasize the feasibility and usefulness of continuous intraoperative nerve monitoring in patients with a non-recurrent laryngeal nerve.

**Abstract:**

Thyroid surgery is associated with a risk of injury to the recurrent laryngeal nerve, especially in the presence of anatomical variants such as a non-recurrent laryngeal nerve (NRLN). Injury to the nerve leads to transient or permanent vocal cord palsy (VCP). A novel method to prevent VCP is continuous intraoperative nerve monitoring (cIONM), but less is known about the applicability of this method in patients with NRLN. The aim of this study was to evaluate our own data regarding feasibility and detailed characteristics of cIONM in NRLN patients. We performed a monocentric retrospective cohort analysis including clinical data and intraoperative nerve monitoring data (measured by Inomed Medizintechnik GmbH, Emmendingen, ‘C2’ and ‘C2 Xplore’ device) of all thyroid surgery patients, showing NRLN between 2014 and 2022. Of 1406 patients who underwent thyroid surgery with cIONM between 2014 and 2022, 12 patients (0.9%) showed NRLN intraoperatively. Notably, cIONM was feasible in eight patients (67%). In all cases the onset latency of the right vagus nerve was shorter (<3.0 ms) than usually expected, suggesting that a short latency might be suitable to distinguish NRLN. None of the patients had a post-operative VCP. Overall, cIONM appears to be feasible and safe in NRLN patients and provides helpful information to prevent VCP.

## 1. Introduction

Thyroid surgery is associated with a risk of injury to the inferior laryngeal nerve, also known as the recurrent laryngeal nerve (RLN). About 1–2% of the human population show an anatomical variant of the right RLN, the so-called non-recurrent laryngeal nerve (NRLN) [1,2]. Various courses of the NLRN are described in the literature, but Type 1 (sometimes called the “descending”, coursing along the superior thyroid pedicle; see Figure 1—course 1), Type 2a (parallel and above the inferior thyroid artery; see Figure 1—course 2) and Type 2b (parallel and below the inferior thyroid artery; see Figure 1—course 3) are the most common variants [1,3,4]. Of note, NLRN is often associated with the presence of an arteria lusoria, an aberrant right subclavian artery arising from the descending aorta, and the absence of the brachiocephalic trunk [5]. Due to the non-recurrent course of the NRLN (see Figure 1), the nerve is at increased risk of an intraoperative injury during thyroid surgery [2,6].

In general, injury to the regular RLN occurs in up to 10% of thyroid surgical patients, leading to a transient or even permanent vocal cord palsy (VCP) depending on the extent of damage [7]. Unilateral VCP goes along with dysphonia, dysphagia and dyspnea, being associated with higher morbidity due to the development of complications such as aspiration pneumonia [8]. A rare but serious issue after thyroid surgery is a bilateral VCP, as affected patients cannot breathe adequately due to an airway obstruction caused by the paralyzed vocal folds [8,9]. Bilateral VCP usually requires a tracheostomy or other airway improvement surgery [10]. The gold standard of thyroid surgery is the visual identification of the RLN, but several publications emphasize the role of an intermittent intraoperative nerve monitoring (IONM) to support the prevention of injury to the RLN and preservation of its functional integrity [11,12]. For IONM, suprathreshold stimulation (1–2 mA) of the vagus nerve and RLN is performed with a hand-held stimulation probe. Surface electrodes on the endotracheal tube, which are positioned at the level of the vocal folds, subsequently register the compound muscle action potentials of the laryngeal muscles [13]. A novel technique to enable real-time evaluation of the nerve’s functionality is continuous intraoperative nerve monitoring (cIONM) [13]. Also, cIONM complements the technique of IONM with repetitive stimulation, which offers the advantage of an immediate feedback in case of imminent traction damage to the RLN [14,15]. Recent publications have stated that cIONM is able to reduce transient and permanent VCP rate more effectively than intermittent IONM, but the practicability and plausibility of cIONM is still a matter of debate [14,16,17,18]. Some authors emphasize the potential role of cIONM in patients with anatomical variants [12,14]. To date, there are limited data reporting the use of cIONM to prevent vocal cord palsy in patients with NRLN. We hypothesized that cIONM may be helpful in NRLN patients to prevent transient and permanent vocal cord palsy. Therefore, the aim of this study was to evaluate our own results as a major thyroid surgery center regarding the applicability and detailed characteristics of cIONM in NRLN patients.

## 2. Materials and Methods

We performed a monocentric retrospective cohort analysis that included clinical data and the IONM data of all thyroid surgery patients showing NRLN between June 2014 and February 2023. The aim of this exploratory pilot study was to evaluate the feasibility and detailed characteristics of CIONM in patients with NRLN. All patients with a preoperative intact vocal cord function who underwent thyroid surgery were eligible for evaluation, regardless of age, sex, or entity of thyroid disease. Preoperative intact vocal cord function was verified by laryngoscopy.

Due to the exploratory character of the study and the rarity of NRLN, no sample-size calculation was performed prior to conducting the study. All data were captured in a pre-formed table and analyzed descriptively. Continuous numeric data are reported as mean and range. Categorical data are presented as the absolute number of all patients with the specified characteristic and as a percentage of all included patients. The reported intraoperative descriptions of NRLN courses were used to assign the patients to the different nerve types, as described by Toniato et al. and Weiand and Mangold [1,3].

All included patients gave written informed consent for their data to be used for research within the EUROCRINE^®^ registry. All procedures were performed in accordance with the ethical standards of the institutional and national research committee, as well as with the 1964 Helsinki declaration and its later amendments or comparable ethical standards.

### Intraoperative Nerve Monitoring

IONM was performed according to the recommendations of the German Surgical Working Group for Endocrinology and the guideline of the International Neural Monitoring Study Group [19,20]. IONM raw data were generated using the ‘C2’ and ‘C2 Xplore’ devices (inomed Medizintechnik GmbH, Emmendingen, Germany). The used device displays the onset latency (time between stimulation and the first peak (negative or positive) of the action potential). IONM raw data were stored and underwent cleaning to ensure data quality with the software tool MIONQA (Mainz IONM Quality Assurance and Analysis Tool, Mainz, Germany; source code: https://ionmreference.net/downloads/mionqa-software/ (accessed on 28 November 2023)) as described before [21]. Before starting the cIONM, a non-continuous IONM of the vagus nerve was performed. During surgery, the vagus nerve, the NRLN and the superior laryngeal nerve were occasionally measured by IONM. For cIONM, repeated stimulations of the right vagus nerve were effected by a so-called ‘delta’ electrode (inomed Medizintechnik GmbH, Emmendingen, Germany), which was placed around the vagus nerve at the start of surgery. The vagus nerve–RLN axis was used in both nerve monitoring procedures, implying the integrity of the vagus nerve and the NRLN. The electromyography (EMG) of the resulting vocal fold contractions were recorded at the level of vocal folds using adhesive laryngeal electrodes with four channels surrounding the whole breathing tube.

The graphs shown in Figures 2–4 and 6 include a separate presentation for left sided (“left”) and right sided (“right”) resections in the case of a bilateral operation. The X-axis indicates the time of registration, while the Y-axis shows the percentage (lower image) and absolute values (upper image) of amplitude (mV, blue line/dots) and the latency (ms, red line/dots) with respect to the selected baseline. The 50% threshold is indicated by the dashed line. Setting of baseline is marked by a green dot; other dots are illustrating IONM stimulation distal to the vagus nerve (pink dots), of the non-recurrent laryngeal nerve (yellow dots) and of the superior laryngeal nerve (orange dots). The solid lines represent stimulation with the vagal electrode while the red and blue dots represent intermittent stimulations with the hand-held stimulation probe. Figures 2–6 were produced with the R-based shiny application MIONQA, (which uses R 4.3.1 and, among others, the packages “ggplot2 3.4.4” and “plotly 4.10.4”.

## 3. Results

Of 1406 patients (of which 69% were female) who underwent thyroid surgery between 2014 and 2023, 12 patients (0.9%) showed NRLN intraoperatively. All of the NRLN patients were female. The mean age of the included patients was 31 years (range 16–57 years). Most of the patients (n = 8, 67%) underwent surgery for a nodular goiter. Two patients (17%) suffered from malignant disease (papillary thyroid carcinoma). Six patients (50%) underwent thyroidectomy, and five patients (42%) had a right lobectomy. Surgery was performed by the same surgeon in all cases, except for one. In all patients, the results of laryngoscopy revealed that vocal fold function was without pathological findings pre- and post-operatively. A short overview of patients’ characteristics is shown in Table 1.

Nerve-related characteristics are shown in Table 2. The operative report revealed that the vagus nerve ran more medially and dorsally to the carotid artery than normally expected in seven patients (58%).

Patients had the following anatomical variants of the laryngeal nerve: four patients (33%) showed a descending NRLN (assigned to Type 1), six patients (50%) had a Type 2a NRLN, and an ascending NRLN (assigned to Type 2b) was found in two patients (17%). In two patients (17%) further branching of the nerve was observed. As an example of the intraoperative anatomy in cases of NRLN, Appendix A shows the intraoperative situation in Patient 8. In Appendix A, no IONM signal can initially be seen via the vagus nerve. After continuing the cranial dissection of the vagus nerve, an early branching of the NRLN (corresponding to Type 1) can be observed.

It was found that cIONM was feasible in eight patients (67%). In almost all patients with a Type 1 NRLN, electrode insertion was avoided to preserve injury to the vagus nerve and the NRLN due to the nerve’s early ramification. In another patient with a Type 2a NRLN and a large goiter, the electrode was also not inserted to preserve injury. The recorded vagus and NRLN signals based on the cIONM of patients who underwent right lobectomy or lymphadenectomy, and of patients who underwent thyroidectomy, are shown in Figure 2 and Figure 3, respectively. We observed no adverse events during cIONM. In Figure 2, Patient 4 and Patient 9 showed a drop of cIONM signal amplitude below the 50% threshold. Patient 4 experienced a rapid loss of signal (between 11:40 and 11:50), which led to a brief surgical pause for signal recovery. After completion of the right-sided surgery, the left side was examined (single IONM signal at the end of surgery). In Figure 3, Patient 3 and Patient 7 showed a drop of the cIONM signal amplitude below the 50% threshold, but the signal recovered rapidly after resection. In all patients shown in Figure 2 and Figure 3, the results of laryngoscopy revealed that vocal fold function was without pathological findings post-operatively.

Results of the non-continuous IONM are shown in Figure 4.

**Figure 2 cancers-16-01007-f002:**
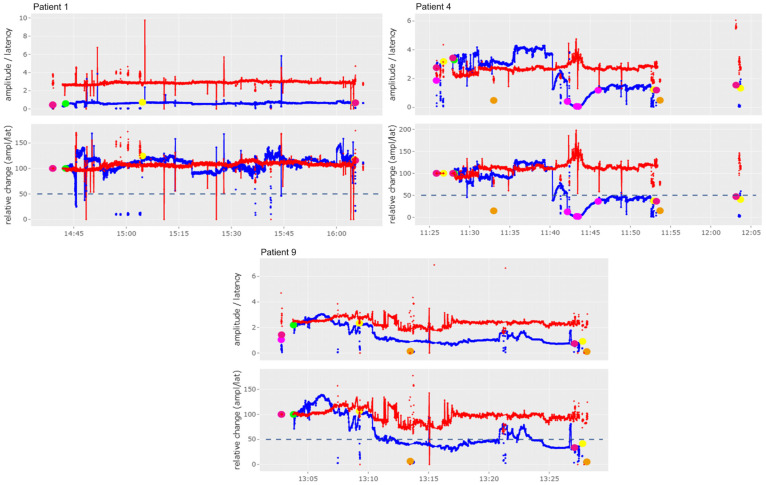
Results of cIONM (continuous intraoperative nerve monitoring) of patients who underwent lymphadenectomy (Patient 1) or right lobectomy (Patient 4 and Patient 9). Detailed description of markings and colors can be found in Section 2 of this manuscript.

**Figure 3 cancers-16-01007-f003:**
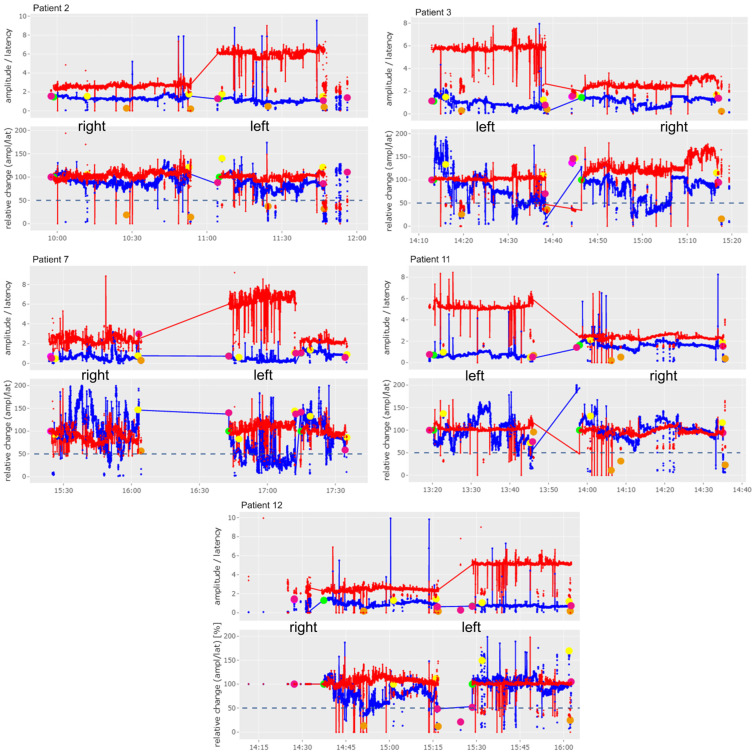
Results of cIONM (continuous intraoperative nerve monitoring) of patients who underwent thyroidectomy (Patient 2, 3, 7, 11 and 12). Detailed description of markings and colors can be found in Section 2 of this manuscript.

**Figure 4 cancers-16-01007-f004:**
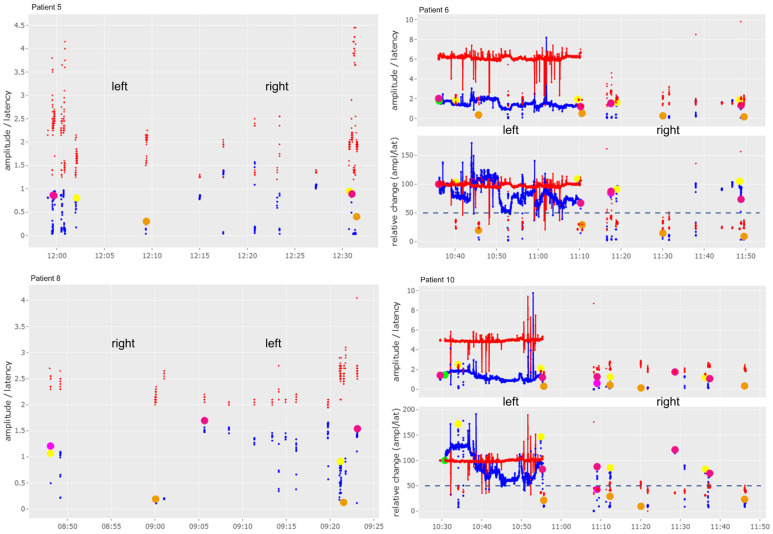
Results of IONM (intraoperative nerve monitoring) of patients who underwent a right lobectomy (Patient 5 and Patient 8) or thyroidectomy (Patient 6 and Patient 10). Detailed description of markings and colors can be found in Section 2 of this manuscript.

### Short Digression: Latency of the Vagus Nerve in Case of NRLN

When considering the EMG results of our study, it is important to note that the device used displays the onset latency, the time between stimulation and the beginning of the action potential. Other devices display the peak latency, the time between stimulation and the first peak (negative or positive) of the action potential. Thus, differences in the expected default values of IONM latencies can be found in the recent literature. In our study, the onset latency of the right vagus nerve was shorter than usually expected. In a previous publication by our research group analyzing IONM data from nearly 2000 patients, we demonstrated that the right vagus nerve has an average onset latency of 4.2 ms [21]. In this study, the right vagus nerve onset latency was shorter than 3.0 ms in all NRLN patients. We assume that an onset latency of less than 3.0 ms might be suitable to distinguish NRLN, which is also underlined by one of our previous publications [21]. When comparing patients with and without NLRN, there is a clear difference in the onset latency of the vagus nerve (see Figure 5). In Figure 5, onset latency was lower than 3.0 ms in all patients with NRLN (green dots), whereas it was lower than 3.0 ms in only four patients without NRLN (red dots). No anatomical variants were reported in the operative reports of these patients. Patients underwent a thyroidectomy (Patient A, C and D) or parathyroidectomy (Patient B). Patient A was a 12-year old male child who underwent surgery due to Grave’s disease. Patient B underwent surgery due to secondary hyperparathyroidism. Patients C and D were both young women who underwent surgery due to a goiter and papillary thyroid carcinoma, respectively. Postoperative laryngoscopy was without signs of vocal cord palsy in all patients. A possible reason for these outlier values may be that a Type 3 non-recurrent laryngeal nerve was unnoticed during surgery, as described in the literature. The results of IONM with these patients are shown in Figure 6.

Some patients without a NRLN showed unusually high latencies of the vagus nerve (>5.0 ms), which could be due to pre-damage of the nerve axis.

**Figure 5 cancers-16-01007-f005:**
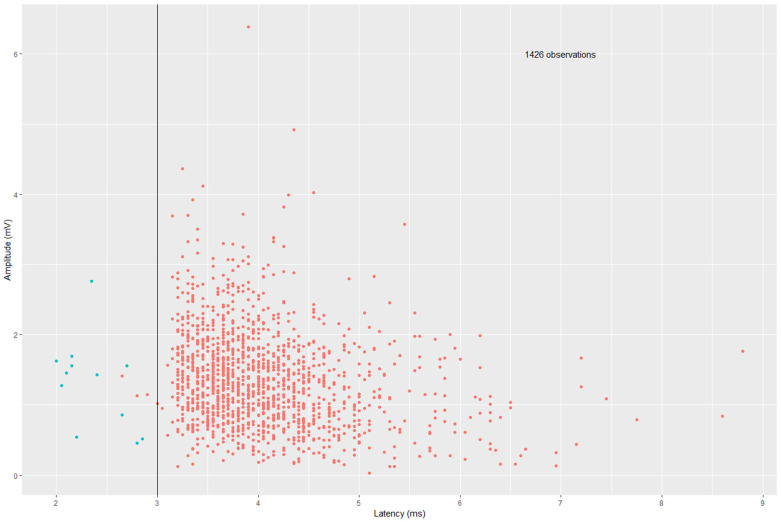
Comparison of vagus nerve electromyography (EMG) results of 1426 observations in patients with (green dots) and without a non-recurrent laryngeal nerve (red dots). The X-axis shows the latency (ms), the Y-axis indicates the amplitude (mV). The black line marks the cut-off of 3.0 ms.

**Figure 6 cancers-16-01007-f006:**
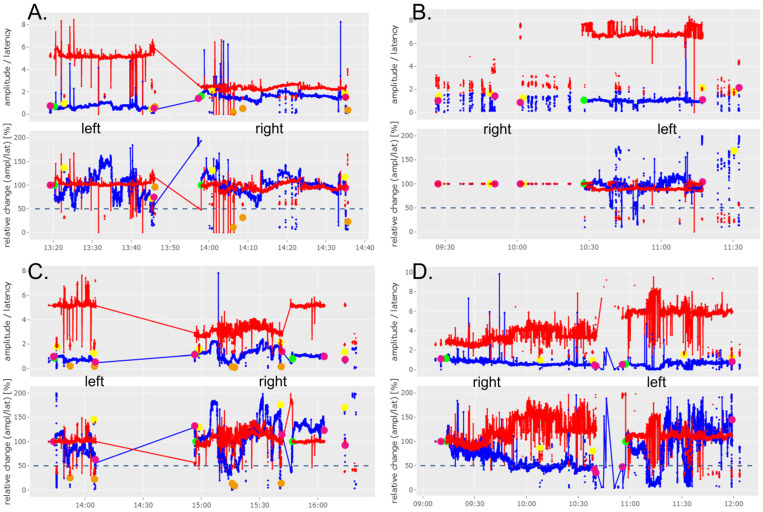
Results of cIONM (continuous intraoperative nerve monitoring) for the four outlier patients ((**A**–**D**); see main text for more information) who had an onset latency below 3.0 ms without existence of a non-recurrent laryngeal nerve. Detailed descriptions of markings and colors can be found in Section 2 of this manuscript.

## 4. Discussion

This is one of the first studies evaluating the feasibility of cIONM in NRLN patients. The results of our study show that cIONM was feasible and safe in almost all of our patients with Type 2 NRLN. In Type 1 NRLN patients, proximal dissection of the vagus nerve was omitted for safety reasons so that no electrode for cIONM could be inserted. A case report about two NRLN patients by Van Slycke et al. revealed that the IONM of the vagus nerve was not possible at all due to the anatomical variants in NRLN [22]. The careful dissection of the internal jugular vein and the common carotid artery is challenging and needs an experienced surgeon, but it is assumable that the approach of identifying the vagus nerve first helps to identify and spare the NRLN. Additionally, the use of IONM of the vagus nerve appears to be able to support the identification of NRLN. In cases of NRLN, it is known that IONM distally of the ramification of the inferior laryngeal nerve does not reveal an IONM signal [23]. Usually, the proximal IONM of the right vagus nerve produces a shorter than normal latency in cases of NRLN. The short latency is suspicious but not probative of the presence of NRLN [21,24]. It is known that damaged nerves can also show an increased latency [25]. The definition of “short” depends on the device used, as onset latencies of less than 3.0 ms and peak latencies of less than 3.5 ms are defined as shorter than usual [19,21]. The recently reported standard values are generally peak latencies [19,26]. It is therefore important to distinguish between onset latency and peak latency. Brauckhoff et al. stated that a peak latency of less than 3.5 ms is able to discriminate NRLN [24]. As described above, our device uses onset latencies [19]. Onset latencies are shorter than peak latencies (the right vagus nerve, for example, has a peak latency of 5.47 ms compared to an onset latency of 4.2 ms [19,21]). Based on our preliminary observations, we advocate a cut-off of 3.0 ms for the onset latencies to discriminate NRLN patients, which is in line with the observed results in this study.

Fortunately, none of our patients suffered from VCP post-operatively, which is a good outcome, since NRLN patients are at high risk of VCP due to the anatomical variants of the nerve [2,6]. It is likely that not only the surgeon’s expertise but also cIONM contributed to this result, indicating that cIONM might be a useful tool in NRLN patients. Several publications emphasize the role of cIONM in reducing transient and permanent VCP rates in thyroid surgery [14,16,17]. In 2020, Cossa et al. reported their experience with cIONM in NRLN patients [27]. The authors evaluated the results of more than 1000 thyroid surgeries, including seven surgeries of NRLN patients. In three NRLN patients, cIONM was used, and none of these patients developed VCP, whereas two out of four NRLN patients in the IONM group developed VCP (one permanent VCP due to neurotmesis and one transient VCP). The authors concluded that cIONM might be a helpful tool to prevent VCP in NRLN patients [27]. Of course, the publication of Cossa et al. is limited due to its very small sample-size [27]. The rarity of NRLN makes it almost impossible to achieve large sample sizes and to conduct randomized-controlled trials, which is also a limitation of our study. Several publications cite NRLN rates of 1–2% in the population, but our study found a rate of less than 1%, which is close to the findings of Cossa et al., who reported seven NRLN patients in 1074 operations (0.6%) [1,2,27]. Other studies report a higher NRLN incidence of up to 6%, indicating local and possibly ethnic differences [28]. The embryological reason for the occurrence of NRLN is still a matter of debate. The inferior laryngeal nerve is formed during the fifth to sixth week of gestation, and is associated with the sixth branchial arch [4]. The aortic arch is created cranial to the larynx at an early gestational stage, but the neck elongates during gestation and the larynx moves cranially [29]. At this point, the inferior laryngeal nerve is already connected to the larynx, and the movement of the larynx causes the loop-like configuration of the right RLN around the right subclavian artery [4]. This also explains why the NRLN is often associated with the occurrence of an aberrant right subclavian artery as the arterial course differs [5]. However, it still remains unclear why all of the NRLN patients do not show an aberrant right subclavian artery [29]. Interestingly, in our study, all of the NRLN patients were female. This observation may be biased by the fact that women are more likely to undergo thyroid surgery than men; to the best of our knowledge, the recent literature does not indicate that NRLN occurs more frequently in women than in men.

Even if the results are limited due to the retrospective character of the study and the monocentric study design, the results are of high clinical relevance and can contribute to the improvement of thyroid surgery. Noteworthy other anatomical variants of the recurrent course with respect to branching and location are known, in addition to the possibility of a non-recurrent course of the nerve [30,31]. Also, cIONM might be able to prevent injury to all variants of the laryngeal nerve due to the real-time evaluation of the vagus nerve/RLN axis. Nevertheless, cIONM usage needs experience, and all surgeons must be aware of cIONM-related technical pitfalls, such as incorrect positioning of the endotracheal electrode or the occurrence of a laryngeal twitch [13,19,21]. In our evaluation, two patients showed a permanent drop of the cIONM signal amplitude during surgery without a clinically apparent VCP after surgery. One might say that cIONM overestimates the occurrence of nerve injury, but it is known that the loss of cIONM signal (LOS) occurs after a multiple combined event (mCE, a combination of a drop of the cIONM signal amplitude below the 50% threshold and an increase in latency of 10%) [15,32]. LOS is associated with nerve injury and the consecutive VCP [15]. Therefore, cIONM constitutes an early warning system to prevent VCP. The safety of cIONM has been demonstrated in a variety of publications [32]. Side effects are rare and usually completely reversible after removal of the stimulation electrode [32,33]. Also, cIONM still offers some research opportunities, as well as the possibility of integrating artificial intelligence (AI). AI might be able to recognize the above-mentioned technical pitfalls and support the surgeon in his assessment of the nerve and its functionality. However, there are no valid research results regarding this idea to date.

## 5. Conclusions

The use of IONM provides helpful information in patients with NRLN and can contribute to the prevention of nerve injury and consecutive VCP. Our data also suggest that an onset latency of less than 3.0 ms might be suitable to distinguish NRLN, which could facilitate the detection of the nerve before the start of dissection. Additionally, cIONM appears to be feasible and safe in patients with NRLN. Furthermore, cIONM bears the possibility of a real-time evaluation of the nerve’s functionality. Future research has to verify the validity and reproducibility of our data.

## Figures and Tables

**Figure 1 cancers-16-01007-f001:**
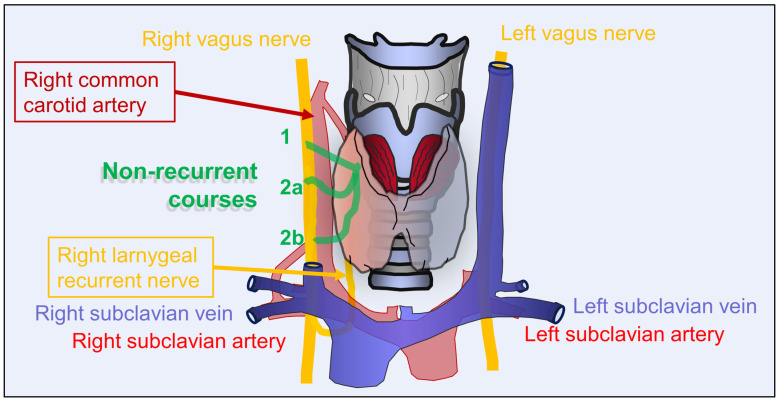
Scheme of the regular course of the right laryngeal recurrent nerve and its common anatomical non-recurrent variants (Type 1, Type 2a, Type 2b; according to Toniato et al., Weiand and Mangold [1,3]). Anatomical variants of the left laryngeal recurrent nerve are extremely rare, which is why they are not shown.

**Table 1 cancers-16-01007-t001:** Overview of included patients. All included patients were female.

No.	Age	Disease	Surgery	Laryngoscopy	
				Preoperative	Postoperative
1	27	PTC	lymphadenectomy	normal	normal
2	40	PTC	thyroidectomy	normal	normal
3	49	multinodular goiter	thyroidectomy	normal	normal
4	20	right nodular goiter	right lobectomy	normal	normal
5	17	right nodular goiter	right lobectomy	normal	normal
6	30	Graves’ disease	thyroidectomy	normal	normal
7	34	right nodular goiter	thyroidectomy	normal	normal
8	38	right nodular goiter	right lobectomy	normal	normal
9	16	right nodular goiter	right lobectomy	normal	normal
10	39	toxic multinodular goiter	thyroidectomy	normal	normal
11	57	Graves’ disease	thyroidectomy	normal	normal
12	38	multinodular goiter	thyroidectomy	normal	normal

PTC = papillary thyroid carcinoma.

**Table 2 cancers-16-01007-t002:** Overview of nerve-related characteristics.

No.	Vagus Nerve Anatomy *	cIONM	Type of NRLN
1	n. s.	yes	2a
2	n. s.	yes	2b
3	medial dorsal	yes	2a
4	medial dorsal	yes	2a
5	medial dorsal	no	1
6	medial dorsal	no	1
7	n. s.	yes	2b
8	medial	no	1
9	n. s.	yes	2a
10	medial	no	2a
11	medial dorsal	yes	2a
12	n. s.	yes	1

* in relation to the common carotid artery. (c)IONM = (continuous) intraoperative nerve monitoring, NRLN = non recurrent laryngeal nerve, n. s. = not specified.

## Data Availability

Data is available from the corresponding author on reasonable request.

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
