# Peer review of "Continuous Intraoperative Nerve Monitoring of a Non-Recurrent Laryngeal Nerve: Real-Life Data of a High-Volume Thyroid Surgery Center"

_cancers, 2024, doi:10.3390/cancers16051007_

Round 1

Reviewer 1 Report

Comments and Suggestions for Authors

Commentary 1

Very small number of patients, especially when taking into account patients with thyroid cancer

Commentary 2

Please explain the difference between IONM and cIONM – I mean why IONM is not enough for the certain surgical procedures.

Commentary 3

Line 110 - All of the NRLN patients were female – what was the percentage of female patients overall?

Commentary 4

Table 1. Lymphadenectomy – what is the risk to the (NR)LN in case of this type of surgery? Or did you mean Lymphadenectomy + thyroidectomy

Commentary 5

Line 149 - In both patients, pausing for recovery did not help to recover the nerve signal. – what was the reason for the loss of signal, and why was it lost if the VCP did not occur? Does it influence the reliability of IONM in any way?

Commentary 6

Line 179 – EMG – please explain the abbreviation

Commentary 7

Line 274 - To the best of our knowledge, recent literature does not indicate that NRLN occurs more frequently in women than in men – can you provide a reference?

Commentary 8

The conclusion section:

cIONM appears to be feasible and safe in patients with NRLN. Furthermore, cIONM bears the possibility of a realtime evaluation of the nerve’s functionality

In my opinion the study itself and even preparation of the manuscript took so much effort that the conclusions provided in the two aforementioned sentences are unsatisfactory. First thing that comes to my mind when thinking about the advantages of continuous monitoring compared to intermittent monitoring is that it is safe and feasible in patients, and that it gives the possibility of realtime evaluation of the values monitored, without the need for time consuming studies and evaluations.

I think that a really interesting information that comes from this article was the fragment saying that it was possible to distinguish between the RLN and NRLN by the right vagus nerve onset latency times. Perhaps this direction could be further explored by the authors?

Author Response

Response: We would like to thank you very much for careful and thorough reading of this manuscript and for the thoughtful comments and constructive suggestions, which help us to improve the quality of this article. All requested changes have been highlighted in yellow. We appreciate your input, your advice and the fast peer review. Please find our point-by-point response below.

Commentary 1: Very small number of patients, especially when taking into account patients with thyroid cancer

Response: Due to the rarity of the NRLN, one of the strengths of our work is that we were able to include data from more than 10 patients.

Commentary 2: Please explain the difference between IONM and cIONM – I mean why IONM is not enough for the certain surgical procedures.

Response: In general, with intermittent IONM the function of the RLN is only sporadically tested, especially at the beginning and the end of a resection, while cIONM provides information about the RLN function during the complete surgery. Since the main cause of nerve damage is related to traction on the RLN during resection, without visible damage to the nerve itself, damage of the RLN is many times not obvious to the surgery with intermittend IONM. In addition, with cIONM the surgeon is enabled to detect impending nerve damage and therefore may be able to prevent it.

Currently, there is no general recommendation to use cIONM intraoperatively, but we have already discussed the advantages of cIONM in a previous publication (see Staubitz et al., Doi: 10.1007/s40136-021-00353-7). We added the difference between IONM and cIONM to the introduction (“For IONM, suprathreshold stimulation (1-2 mA) of the vagus nerve and RLN is per-formed with a hand-held stimulation probe. Surface electrodes on the endotracheal tube, which are positioned at the level of the vocal folds, subsequently register the compound muscle action potentials of the laryngeal muscles. […] cIONM complements the technique of IONM with repetitive stimulation, which offers the advantage of an immediate feedback in case of an imminent traction damage to the RLN.” (line 63-66 and line 69-71)

Commentary 3: Line 110 - All of the NRLN patients were female – what was the percentage of female patients overall?

Response: The overall percentage of females was 69% (added to line 129).

Commentary 4: Table 1. Lymphadenectomy – what is the risk to the (NR)LN in case of this type of surgery? Or did you mean Lymphadenectomy + thyroidectomy

Response: The patient had a lymph node recurrence of a papillary thyroid carcinoma that had previously been treated by thyroidectomy. Due to revision surgery, it must be assumed that the risk to the nerve was even higher than with the primary operation.

Commentary 5: Line 149 - In both patients, pausing for recovery did not help to recover the nerve signal. – what was the reason for the loss of signal, and why was it lost if the VCP did not occur? Does it influence the reliability of IONM in any way?

Response: The drop of signal is most likely be caused by traction on the RLN during elevation of the thyroid lobe that was necessary for dissection. We addressed this "overestimation" of nerve damage in the discussion: “In our evaluation, two patients showed a permanent drop of the cIONM signal ampli-tude during surgery without a clinically apparent VCP after surgery. One might say that cIONM overestimates the occurrence of nerve injury, but it is known that the loss of cIONM signal (LOS) occurs after a multiple combined event (mCE, a combination of a drop of the cIONM signal amplitude below the 50% threshold and an increase of latency by 10%)15,32. LOS is associated with nerve injury and the consecutive VCP15. Therefore, cIONM constitutes an early warning system to prevent VCP.” (line 278-285)

Commentary 6: Line 179 – EMG – please explain the abbreviation

Response: Added to the manuscript and to abbreviations (electromyography (EMG)).

Commentary 7: Line 274 - To the best of our knowledge, recent literature does not indicate that NRLN occurs more frequently in women than in men – can you provide a reference?

Response: Other publications do not differentiate between male and female NRLN patients. Our observation has therefore not yet been studied in detail in other studies, but we assume that differences between men and women would have been described previously in anatomical studies (for example Weiand et al., Doi: 10.1007/s00104-003-0776-6). In addition, from personal experience with patients operated before the study period we know, that non-recurrent laryngeal nerves occur in men too.

Commentary 8: The conclusion section:

cIONM appears to be feasible and safe in patients with NRLN. Furthermore, cIONM bears the possibility of a realtime evaluation of the nerve’s functionality

In my opinion the study itself and even preparation of the manuscript took so much effort that the conclusions provided in the two aforementioned sentences are unsatisfactory. First thing that comes to my mind when thinking about the advantages of continuous monitoring compared to intermittent monitoring is that it is safe and feasible in patients, and that it gives the possibility of realtime evaluation of the values monitored, without the need for time consuming studies and evaluations.

I think that a really interesting information that comes from this article was the fragment saying that it was possible to distinguish between the RLN and NRLN by the right vagus nerve onset latency times. Perhaps this direction could be further explored by the authors?

Response: Thank you for this important suggestions! We revised the conclusion and the abstract of our manuscript. We added “Our data also suggest that an onset latency of less than 3.0 ms might be suitable to distinguish NRLN, which could facilitate the detection of the nerve before the start of dissection” to the conclusion (line 293-295) and “In all cases the onset latency of the right vagus nerve was shorter (<3.0 ms) than usually expected suggesting that a short latency might be suitable to distinguish NRLN” to the abstract.  

Reviewer 2 Report

Comments and Suggestions for Authors

The argument treated by the Authors in the manuscript entitled “Continuous intraoperative nerve monitoring of a non-recurrent laryngeal nerve: real-life data of a high-volume thyroid surgery center” is of great interest for the medical community.

The Authors have reported their personal experience based on 1406 cases. Of these, the diagnosis of NRLN is reported in 0.9% of studied population.

I will suggest the Authors to discuss the current techniques to monitor the laryngeal nerve also in accordance to different devices and current technologies, with emphasis on artificial intelligence.

Is the artificial intelligence currently used for the definition of anatomical variations of the laryngeal nerve in the preoperative or intraoperative setting?

The discussion is of good quality and underlines the limits of a retrospective and monocentric study.

Thank you for your proposed manuscript.

Comments on the Quality of English Language

The present language quality is good enough and needs minor improvements.

Author Response

The argument treated by the Authors in the manuscript entitled “Continuous intraoperative nerve monitoring of a non-recurrent laryngeal nerve: real-life data of a high-volume thyroid surgery center” is of great interest for the medical community.

The Authors have reported their personal experience based on 1406 cases. Of these, the diagnosis of NRLN is reported in 0.9% of studied population.

I will suggest the Authors to discuss the current techniques to monitor the laryngeal nerve also in accordance to different devices and current technologies, with emphasis on artificial intelligence.

Is the artificial intelligence currently used for the definition of anatomical variations of the laryngeal nerve in the preoperative or intraoperative setting?

The discussion is of good quality and underlines the limits of a retrospective and monocentric study.

Thank you for your proposed manuscript.

Response: Thank you for this interesting suggestion! We use AI as part of our preoperative diagnostics to evaluate the dignity of thyroid nodules, but so far, we haven’t use AI intraoperatively. To date, data on intraoperative AI usage is sparse and the main focus is on assessing the dignity of thyroid nodules and the existence of lymph node metastases (see for example Taha et al., Doi: 10.1016/j.amjsurg.2023.11.019). Nevertheless, AI could help to identify NRLN by evaluating the values obtained by (c)IONM, which is why we added your suggestion to the discussion of our manuscript: “cIONM still offers some research opportunities and also the possibility of integrating artificial intelligence (AI). AI might be able to recognize the above-mentioned technical pitfalls and support the surgeon in his assessment of the nerve and its functionality. However, there are no valid research results to date” (line 287-290)

Reviewer 3 Report

Comments and Suggestions for Authors

            COMMENTS  

The manuscript titled “Continuous Intraoperative Nerve Monitoring of a Non-Recurrent Laryngeal Nerve: Real-Life Data of a High-Volume Thyroid Surgery Center” of Ann-Kathrin  Ledereret al., reports an investigation about the usefulness of continuous intraoperative nerve monitoring (cIONM) to prevent vocal cord palsy  (VCP) in patients with a non-recurrent laryngeal nerve (NRLN).

The foundation of this study lies in two evidences. Firstly, thyroid surgery is associated with risk of injury to recurrent laryngeal nerve, especially in presence of anatomical variations, namely NRLN. Secondly, injury to the nerve leads to transient or permanent VCP.

The aim of this investigation was to evaluate the feasibility of a novel method, named as cIONM, to prevent VCP in NRLN patients. By retrospective cohort analysis, this monocentric investigation examined clinical and cIONM data of all thyroid surgery patients (n. 1406) showing NRLN (n. 12) between 2014 and 2022.

To perform this study, cIONM were measured by Inomed Medizintechnik GmbH, Emmendingen, ‘C2’ and ‘C2 Xplore’ device.

Four were the main results of this study.

Firstly, 12 patients (0.9%) showed NRLN intraoperatively.

Secondly, cIONM was feasible in 8 patients (67%).

Thirdly, in all cases the onset latency of the right vagus nerve was shorter (<3.5 ms) than usually expected.

Lastly, none of the patients had a postoperative VCP.

In conclusion, cIONM appears to be feasible and safe in NRLN patients, and provides helpful information to prevent VCP.

Simple Summary:

Simple Summary section is adequately describing this study. However,

Minor:

Lines 13-15: this sentence should be revised. I suggest to authors to specify that this study is “a retrospective case series trial”.

Abstract:

Abstract section is adequately describing this study.

Introduction:     

This section is adequately describing the aims of study.

Materials and Methods:           

This section provides sufficient information.

Results:

This section provides detailed information.

Discussion:

The comments of discussion are appropriate for this investigation.

Conclusions:      

The conclusions are relevant.

Tables, Figures and Video 1 (supplementary materials): give a helpful visual representation of study.

References:

References are adequate.

Decision:

This study may be accepted for publication after minor revision.

Author Response

The manuscript titled “Continuous Intraoperative Nerve Monitoring of a Non-Recurrent Laryngeal Nerve: Real-Life Data of a High-Volume Thyroid Surgery Center” of Ann-Kathrin Lederer et al., reports an investigation about the usefulness of continuous intraoperative nerve monitoring (cIONM) to prevent vocal cord palsy (VCP) in patients with a non-recurrent laryngeal nerve (NRLN). The foundation of this study lies in two evidences. Firstly, thyroid surgery is associated with risk of injury to recurrent laryngeal nerve, especially in presence of anatomical variations, namely NRLN. Secondly, injury to the nerve leads to transient or permanent VCP. The aim of this investigation was to evaluate the feasibility of a novel method, named as cIONM, to prevent VCP in NRLN patients. By retrospective cohort analysis, this monocentric investigation examined clinical and cIONM data of all thyroid surgery patients (n. 1406) showing NRLN (n. 12) between 2014 and 2022. To perform this study, cIONM were measured by Inomed Medizintechnik GmbH, Emmendingen, ‘C2’ and ‘C2 Xplore’ device.

Four were the main results of this study.

Firstly, 12 patients (0.9%) showed NRLN intraoperatively.

Secondly, cIONM was feasible in 8 patients (67%).

Thirdly, in all cases the onset latency of the right vagus nerve was shorter (<3.5 ms) than usually expected.

Lastly, none of the patients had a postoperative VCP.

In conclusion, cIONM appears to be feasible and safe in NRLN patients, and provides helpful information to prevent VCP.

Simple Summary: Simple Summary section is adequately describing this study. However,

Minor:  Lines 13-15: this sentence should be revised. I suggest to authors to specify that this study is “a retrospective case series trial”.

Response: Thank you for this suggestion! The simple summary was revised.

Abstract: Abstract section is adequately describing this study.

Introduction: This section is adequately describing the aims of study.

Materials and Methods: This section provides sufficient information.

Results: This section provides detailed information.

Discussion: The comments of discussion are appropriate for this investigation.

Conclusions: The conclusions are relevant

Tables, Figures and Video 1 (supplementary materials): give a helpful visual representation of study.

References: References are adequate.

Decision: This study may be accepted for publication after minor revision.

Response: Overall, we would like to thank you very much for careful and thorough reading of this manuscript! We are very pleased that you liked our manuscript.

Reviewer 4 Report

Comments and Suggestions for Authors

This paper details a case of NRLN using cIONM among many thyroid surgery cases. Since NRLN cases themselves are very rare, the content of the paper is considered valuable. The conclusion that the response to cIONM varies depending on the type of NRLN is very interesting and will be of great value to physicians who use cIONM in the future. 

The paper is very well organized and appropriate.

The only modification is that the purpose of the paper is not clear from the objectives stated in the abstract and introduction sections. As the author described "the aim of this study was to evaluate our own results as a major thyroid surgery center regarding the applicability of cIONM in NRLN patients",  I have a kind of the impression that it would be written about feasibility of cIONM in NRLN.  I guess it can be easier to understand if it was written in such as "the detailed characteristics of NRLN with cIONM."

Also, the detailed description of Figures is longish, and the small font makes it somewhat difficult to read through. It would be better to give an overview of each Fiuger in the main text and simplify the explanatory text.

Author Response

This paper details a case of NRLN using cIONM among many thyroid surgery cases. Since NRLN cases themselves are very rare, the content of the paper is considered valuable. The conclusion that the response to cIONM varies depending on the type of NRLN is very interesting and will be of great value to physicians who use cIONM in the future.

The paper is very well organized and appropriate.

Response: We would like to thank you very much for careful and thorough reading of this manuscript and for the thoughtful comments and constructive suggestions, which help us to improve the quality of this article. All requested changes have been highlighted in yellow. Please find our point-by-point response below.

The only modification is that the purpose of the paper is not clear from the objectives stated in the abstract and introduction sections. As the author described "the aim of this study was to evaluate our own results as a major thyroid surgery center regarding the applicability of cIONM in NRLN patients", I have a kind of the impression that it would be written about feasibility of cIONM in NRLN.  I guess it can be easier to understand if it was written in such as "the detailed characteristics of NRLN with cIONM."

Response: We revised the aim to “Therefore, the aim of this study was to evaluate our own results as a major thyroid surgery center regarding the applicability and detailed characteristics of cIONM in NRLN patients” (line 76-78) and “The aim of this exploratory pilot study was to evaluate the feasibility and detailed characteristics of CIONM in patients with NRLN” (line 82-83). Additionally, we revised the abstract according to your suggestion (“The aim of this study was to evaluate our own data regarding feasibility and detailed characteristics of cIONM in NRLN patients”).

Also, the detailed description of Figures is longish, and the small font makes it somewhat difficult to read through. It would be better to give an overview of each Fiuger in the main text and simplify the explanatory text.

Response: The descriptions have been shortened. We moved the methods of our figures to the methods section (line 115-127) and added “Detailed description of markings can be found in the methods section of this manuscript” to the legends. The results are now mentioned in the results section of the manuscript (Figure 2 in line 158-160, Figure 3 in line 162-164, Figure 5 and Figure 6 in line 194-206).

Round 2

Reviewer 1 Report

Comments and Suggestions for Authors

All my commentaries have been satisfactorily replied to. I now consider the manuscript to be ready for publication.